# Impact of Clinical Pharmacy Expansion within a Rural Federally Qualified Health Center through Implementation of Pharmacist-Led Medicare Annual Wellness Visits

**DOI:** 10.3390/pharmacy10060160

**Published:** 2022-11-29

**Authors:** Carrington Royals, Reagan K. Barfield, Mary Francis Newman, Lori Mor, Tammy H. Cummings, P. Brandon Bookstaver

**Affiliations:** 1Tandem Health Inc., 1278 N. Lafayette Dr., Sumter, SC 29150, USA; 2Department of Clinical Pharmacy and Outcomes Sciences, University of South Carolina College of Pharmacy, 715 Sumter St., Columbia, SC 29208, USA; 3University of South Carolina School of Medicine – Sumter Family Medicine Residency Program, 129 N. Washington St., Sumter, SC 29150, USA

**Keywords:** medicare annual wellness visits, federally qualified health center, clinical pharmacy, service implementation

## Abstract

Medicare Annual Wellness Visits (AWVs) are annual appointments with the primary care team to prepare personalized prevention plans and focus on gaps in care. Although beneficial, AWVs are often difficult for providers to schedule and complete due to the increased time commitments compared to other visits. The purpose of this study was to assess the clinical, economic and patient-level value of newly implemented pharmacist-led AWVs within a rural Federally Qualified Health Center (FQHC). This retrospective, cohort study included patients who completed an AWV between 1 October 2021, and 14 February 2022. The primary objective was to compare the per clinician rate of completed AWVs between pharmacists and providers. The secondary objectives were to compare revenue generated, interventions made, and patient satisfaction between pharmacist- and provider-led AWVs. During the study period, nine providers completed 139 AWVs (15.4/provider) and two pharmacists completed 116 AWVs (58/pharmacist). Proportions of interventions ordered among those due in eligible patients were similar between pharmacists and providers (47.6% vs. 44.5%; *p* = 0.356). Patient satisfaction was overall positive with no difference between groups. Pharmacist-led AWVs increased completion of AWVs by 83% over a 20-week period, including significantly more initial, compared to subsequent, AWVs than providers. Sustainability of pharmacist-led AWVs at this FQHC is supported by study outcomes.

## 1. Background

According to the U.S. Census Bureau 2022 report, more than 54 million adults ages 65 years and older live in the United States, accounting for 16.9% of the nation’s population [1]. As the United States population continues to age, emphasis is being placed on preventative medicine. In 2011, as part of the Patient Protection and Affordable Care Act, Medicare initiated the Annual Wellness Visit (AWV) program for patients covered by Medicare Part B [2]. Beneficiaries are eligible for an initial AWV followed by subsequent AWVs annually under Medicare Part B coverage at no out-of-pocket cost. These visits allow a dedicated time in the outpatient setting to focus on gaps in screening, vaccines, and overall preventive care opportunities [3]. AWVs are associated with a 5.7% reduction in adjusted total healthcare costs and a significant increase in completion of health screenings [4]. Despite these incentives, only 24% of eligible beneficiaries received an AWV in 2017, up from 7.5% in 2011 [5]. Utilization of AWVs is even lower among racial minority beneficiaries and those living in a rural setting [6,7]. A barrier to performing AWVs is the lack of time and scheduling inflexibility among many providers. AWVs on average require 30–60 min per patient not including the preparation time required prior to the visit, making them difficult to schedule and complete [8]. This presents an opportunity to leverage the skills of ambulatory care trained clinical pharmacists to enhance availability and completion of these visits. Although this is becoming a more widespread practice for pharmacists, Federally Qualified Health Centers (FQHCs), community health centers that receive funding from the Health Resources and Services Administration (HRSA) Health Center Program, are potentially underutilizing pharmacists for this service [9,10]. Clinical pharmacy services in FQHCs have often been limited to providing medication therapy management and a few other indirect and direct patient care services, but AWVs provide an opportunity to expand revenue-generating services offered in these community-based settings [11,12,13,14]. A recent review of the literature of pharmacist practice within FQHCs by Rodis and colleagues concluded pharmacist-directed services are expanding, but more research is needed to explore scope of practice and financial sustainability of these services in FQHCs [11].

Tandem Health is an FQHC in rural South Carolina and includes three practice sites where AWVs are currently being conducted: an adult medicine clinic and two-family medicine clinics. The clinics employ seven physicians, four nurse practitioners, and 12 Family Medicine medical residents in partnership with the University of South Carolina School of Medicine. Tandem Health has two fully staffed in-house pharmacies with one clinical pharmacist and one post-graduate year one (PGY1) pharmacy resident, in partnership with the University of South Carolina College of Pharmacy, embedded in the clinic areas. Prior to this study, AWVs were completed solely by providers, but lacked routine scheduling due to aforementioned challenges. This led to increased gaps in care, incomplete preventive screening for eligible patients, and a missed revenue opportunity with over 1700 outstanding AWVs to be completed. The state of South Carolina does not explicitly define pharmacists’ role in the ambulatory care setting but does not hinder collaborative practice agreements (CPAs) between pharmacists and physicians [15]. The purpose of this study was to assess the clinical, economic and patient-level value of newly implemented pharmacist-led AWVs within a rural FQHC.

## 2. Methods

This was a single-centered, retrospective cohort study conducted at Tandem Health, an FQHC located in Sumter, South Carolina. Patients included in this study were established patients of Tandem Health with Medicare Part B coverage who received either an initial AWV or subsequent AWV at the study site between 1 October 2021 and 14 February 2022. Patients were excluded from the study if they refused an AWV or received an AWV within the last year, thus ineligible for an additional visit during the study period. AWVs conducted by providers during the same period served as the comparator group.

Patients meeting inclusion criteria were identified through the electronic health record, Athena^®^. Data collection was completed accessing Athena^®^ and using a Microsoft Excel^®^ data collection form. Patients were assigned a specific study ID and all Protected Health Information was blinded and contained within a password-protected spreadsheet. The cooperating Institutional Review Board gave exempt approval.

The primary objective of this study was to compare the per-clinician rate of AWVs completed by providers to that completed by clinical pharmacists. The secondary objectives were to evaluate the economic and clinical value of pharmacist-led AWVs by determining revenue generated and interventions recommended and ordered, compared to provider conducted AWVs. Chart reviews were conducted to determine interventions patients were due for at the time of the time of their AWV and if these screenings and vaccinations were ordered at the time of the visit. Economic impact was calculated as total revenue generated through the visits themselves and incentivized quality revenue for completing AWVs for priority patients. Total incentive revenue was generated from reports provided by the study site’s Quality Department at the end of the study period. Additionally, patient satisfaction was gauged by survey administered via telephone calls following their AWV. The survey tool consisted of eight questions used to determine satisfaction with scheduling, ancillary staff, clinician knowledge, duration of appointment, and opportunity to ask questions. Patients were asked to answer all questions based on a Likert scale (1 = very unsatisfied to 5 = very satisfied). Providers and nursing staff were surveyed anonymously via written response during the study timeframe to understand their attitudes toward AWVs. These results were only reported for descriptive purposes and context in the application of clinical pharmacists conducting AWV. Descriptive statistics were used to summarize data. Student *t*-test or chi-square tests were used to compare outcomes between groups where applicable.

## 3. Results

Patients seen by providers or pharmacists during the study period had similar baseline characteristics (Table 1). Majority of patients were female, African American or Black, and over 65 years of age. Patients seen by a pharmacist reported more current active medications (9.4 ± 4.4) compared to those seen by a provider (7.6 ± 4.5) (*p* = 0.0067). 

Nine providers completed 139 visits (15.4 per clinician) and two pharmacists completed 116 visits (58 per clinician). Of the visits conducted by pharmacists, 50% were initial AWVs compared to 6.5% conducted by providers (*p* < 0.0001). Pharmacists increased the amount of AWVs completed at the center by 83.5%. Translated into revenue, providers generated $31,682.27 from visits alone and $2626.67 from incentive revenue. Pharmacists generated $26,439.88 from visit reimbursement and $3798.33 from incentive revenue (Table 2).

Proportions of health maintenance screenings ordered compared to screenings that were due in eligible patients were greater among pharmacists when compared to providers (61.3% of 142 screenings due vs. 66.2% of 139 screenings due, *p* < 0.0001). Proportions of vaccines ordered compared to vaccines the patient was eligible for were similar among providers and pharmacists with providers ordering 37.7% and pharmacists ordering 37.8% of vaccines needed (*p* = 0.968). The most common reason for both groups as to why screenings and vaccines were not ordered when due was that the patient declined the service at that time. Further breakdown of screenings and vaccines are illustrated in Appendix A. 

Pharmacists completed visits in an average of 36 ± 15.3 min compared to providers and nursing staff completing visits in an average of 42 ± 13.5 min (*p* = 0.069). When surveyed, patients consistently stated they were satisfied or very satisfied with the duration of their appointments in both groups. In addition to the appointment duration, patients were overall satisfied or very satisfied with the services rendered by both providers and pharmacists (Table 3). 

When surveyed, nursing staff agreed or strongly agreed they would recommend patients receive their AWVs every year and saw a clear benefit for patients to complete these visits. Providers also agreed they would recommend their patients receive AWVs yearly but were neutral on seeing a clear benefit of the visits. Both providers and nursing staff agreed that of the visits they complete at the clinic most frequently, AWVs were among their least favorite to conduct (Table 4).

## 4. Discussion

This study demonstrates implementation of pharmacist-led AWVs in a rural FQHC allows for increased completion rates of quality visits. Although pharmacist-led AWVs is not a novel concept, data describing pharmacy expansion at a rural FQHC are currently limited. Our results align with and build upon the benefits of pharmacist-led AWVs demonstrated in an Accountable Care Organization (ACO) within an FQHC in Arizona [16]. 

Alhossan and colleagues demonstrated the benefits of pharmacist-led AWVs but did not directly compare the pharmacist-led visits to provider-led visits as was done in our study. By using provider visits as the comparator group, we were able to distinguish the difference between standard of care for completing AWVs within the FQHC and the subsequent results of implementing pharmacy-led AWVs. Pharmacists increased AWVs by 83% within a four-and-a-half-month period while also completing their existing duties and responsibilities. Not only were visits increased overall, but nearly 6.5 times as many initial AWVs were completed by pharmacists. This demonstrates that by allowing more clinicians to complete visits, more patients are being reached within the FQHC who otherwise may not have received the AWV currently recommended and reimbursed annually under Medicare Part B. The value of completing an initial visit for a patient may be multiplied by possibly prompting the desire for future recurring AWVs.

Our finding that pharmacists provide the same level of care as their provider colleagues in terms of screenings and vaccinations offered to patients is echoed in current literature [17]. In both groups, a low proportion of eligible screenings and vaccines were ordered for patients. A possible explanation for this can be the socioeconomic barriers among patients the FQHC serves. Many patients struggle with transportation which could have been a barrier to receive preventative health screenings, which are primarily offered at the local hospital or imaging center. This study did not examine receipt of screenings so it remains unclear patient reasoning for declining screening and whether those ordered were fulfilled. Other reasons for patients not receiving vaccinations could have been vaccine hesitancy and potential cost. Education was provided to patients during each visit, however it is difficult to overcome this longstanding barrier during one appointment. Shen and colleagues demonstrated that patients who received an AWV had higher immunization rates for influenza vaccines compared to those who had not received an AWV (64% vs. 44%, respectively) [18]. These results were supported by a similar study by Tao and colleagues in a 2018 which found 63.8% of patients who had an AWV received their annual influenza vaccine, compared to 41.6% who did not [19]. These results are likely due to having a designated time to discuss preventative healthcare and the benefits of vaccinations. Cost is another issue for certain vaccinations, especially Shingrix™ (zoster vaccine recombinant, adjuvanted). Many of the patients within the FQHC do not have Medicare Part D and would therefore require large, out-of-pocket costs to receive the vaccination. We also encountered many patients with Medicare Part D who state they were unable to afford the vaccine copay and declined the Shingrix™ vaccine. Shen and colleagues later analyzed predictors of vaccinations among Medicare beneficiaries and found that patients who were dual-eligible for Medicare and Medicaid were most likely to receive the Shingrix™ vaccine due to increased affordability [20]. Patient barriers to receiving the recommended health screenings and vaccinations were valid, and the study site is working toward reducing these barriers for patients.

Pharmacists in the outpatient setting are continually finding ways to increase revenue in order to justify positions [21]. Many times, pharmacists are unable to directly bill for their services and may be required to conduct joint visits with providers in order to increase revenue indirectly [22]. Pharmacist-led AWVs provide a direct billing opportunity for pharmacists outside of provider encounters. Park and colleagues completed a financial analysis which concluded AWVs were an option for providing sustainable revenue that allowed for pharmacy expansion. The analysis found that a total of 1070 AWVs per year are required to support a pharmacist’s salary [8]. The area where the analysis was completed had a lower reimbursement for AWVs than our study location, which resulted in the higher threshold of AWVs to be completed per year for cost justification. Pierce and colleagues described the expansion of a PGY-2 pharmacy residency program which was funded on revenue from AWVs completed by the residents within the clinic. While resident salaries are generally half of a full time pharmacist’s salary, these authors demonstrated reimbursement from 407 AWVs completed by pharmacy residents throughout the year fully funded the additional pharmacy resident position [23]. Our pharmacist and pharmacy resident generated additional revenue for the FQHC that otherwise may not have been incurred. It was determined at our FQHC that in order to support an additional pharmacist’s salary, 505 AWVs per year must be completed. Incoming pharmacy residents are also now expected to complete visits as part of their longitudinal curriculum. Pharmacists are a cost-effective method for providing an additional resource for patients to receive their AWVs. According to the online database, Salary, the median Medical Doctor’s salary in South Carolina for 2022 is $192,800, or $100 per hour on a 48-week calendar [24]. Generally, in South Carolina clinical pharmacists’ salaries range from $100,000 to $115,000 in the ambulatory care setting, or $52–$60 per hour on a 48-week calendar. By pharmacists performing these hour-long visits, the clinic is adding to the profit from AWVs which adds to the return on investment for a clinical pharmacist. 

Overall patients were very satisfied with both pharmacy and provider visits. This is supported by data from previous studies as well. Following implementation of pharmacist-led AWVs at an academic medical center in North Carolina, Sherrill and colleagues found that patients were very satisfied with the pharmacist and felt comfortable with the pharmacist as providers of the service. They also found patients who received more interventions during the visit were in general more satisfied following the visit [25]. Shin and colleagues analyzed satisfaction with clinical pharmacists within an FQHC and how demographics affected satisfaction ratings. This study found no difference between demographic data and patient satisfaction except for race in which African American patients were significantly more satisfied than Hispanic patients [26]. Although our study did not assess the connection between interventions made or demographic data and patient satisfaction, our patients were also very satisfied with the clinical pharmacists and there was no difference between the groups. 

It was apparent by our survey that providers generally dislike conducting AWVs, likely for multiple reasons, including lack of time. These negative attitudes toward the visits could be an additional explanation for the low completion rates of visits prior to implementation of pharmacist-led AWVs. Wilson and colleagues surveyed providers to gauge satisfaction with pharmacists completing the AWVs. Providers strongly disagreed that they would prefer to do the visits themselves and strongly agreed that patients benefited from a pharmacist-led AWV [27]. Although not formally analyzed in our study, oral feedback from providers within the clinic demonstrated satisfaction with the service provided by clinical pharmacists. Kadakia and colleagues interviewed providers within a small FQHC to summarize factors before implementing a new pharmacy-led service. The major themes this study synthesized were communication, pre-specified roles, logistics, and communication of the service to the community [28]. Our team thoroughly vetted the implementation of pharmacy-led AWVs using these domains prior to our study and this could have played a role in the high provider buy-in we saw within our FQHC and subsequent success of the service.

Several study limitations exist. Beginning with the process of AWVs, a CPA was not developed for the implementation of pharmacist-led AWVs which limited the scope of pharmacists within the visits by restricting them from making medication interventions during the visits. CPAs between pharmacists and physicians have been proven to improve patient management in many settings including FQHCs [29,30]. A CPA is currently being developed and approved for future pharmacy-led AWVs to allow pharmacists more opportunities to improve patient care. Additionally, pharmacists were unable to order labs during the visits and required physicians to sign off on all orders for preventative screenings and vaccines. This limited the timing of vaccines that could have been administered through the center’s on-site pharmacy such as Shingrix™ and Boostrix^®^, which could partially explain the low percentage of vaccines ordered during visits. Our study did not gather data on the number of patients who refused an AWV offered by a pharmacist or the reasons for declining. Other limitations were noted with the survey methodology. Patient satisfaction surveys were administered by the clinical pharmacist who did not complete the visit for the patient. The surveys were not administered at a specific time following the visit, but rather as the clinical pharmacists had time to complete them while completing their other clinic duties. This could have introduced recall bias into the results from the patient satisfaction surveys. Patients were not asked about their willingness to see the clinical pharmacist for their subsequent AWVs which could have provided additional insight into the sustainability of the service. Providers were not formally surveyed following service implementation, however; providers informally expressed their satisfaction with the service and stated it added minimal work to their current workload.

Based on the results of our study, the pharmacy-led AWV model was deemed a cost-effective solution for providing AWVs to more patients without compromising the quality of the visit. The decision was made to continue offering pharmacist-led AWVs to all eligible patients within the FQHC by expanding employment with an additional FTE for a clinical pharmacist. As demonstrated in previous studies, AWVs serve as a sustainable model for growth of pharmacy services and could potentially lead to more clinical pharmacists or additional pharmacy residents in the future.

## 5. Conclusions

Pharmacists increased completion rate of AWVs by 83% within the study timeframe. More patients were able to be seen and therefore, more screenings and vaccines were completed compared to prior standard of care. In addition to the volume of visits completed, the FQHC benefited from the addition of pharmacist-led AWVs by additional revenue exceeding $26,000 and improved quality compliance which also has a large return on investment, both directly and indirectly. It was beneficial to have additional clinicians seeing patients, assessing medication use, and providing medication recommendations where appropriate, but further evaluations are needed to determine the long-term impact on patients’ overall health.

## Figures and Tables

**Table 1 pharmacy-10-00160-t001:** Baseline Characteristics of Patients Who Received an Annual Wellness Visit.

Baseline Characteristics	Overall Patients; *N* = 255	Provider Patients; *N* = 139	Pharmacists Patients; *N* = 116	*p*-Value
**Rendering Clinician**				
Pharmacist; *n* (%)	116 (45.5)	-	-	-
Provider; *n* (%)	139 (54.5)	-	-	-
**Gender**				
Female; *n* (%)	159 (62.4)	78 (56.1)	81 (69.8)	0.0244
Male; *n* (%)	96 (37.6)	61 (43.9)	35 (30.2)	
**Age**				
<65 years; *n* (%)	52 (20.4)	26 (18.7)	26 (22.4)	0.1309
65–70 years; *n* (%)	100 (39.2)	49 (35.3)	51 (44.0)	
>70 years; *n* (%)	103 (40.4)	64 (46.0)	39 (33.6)	
**Race**				
African American or Black; *n* (%)	171 (67.0)	86 (61.9)	85 (73.3)	0.1257 *
White; *n* (%)	81 (31.8)	51 (36.7)	30 (25.9)	
Other; *n* (%)	3 (1.2)	2 (1.4)	1 (0.8)	
**Medical Background**				
Scheduled Medications; Mean ± SD	8.4 ± 4.4	7.6 ± 4.5	9.4 ± 4.4	0.0067
Chronic Comorbidities; Mean ± SD	7.0 ± 3.1	7.5 ± 3.2	6.4 ± 3.2	

* Fisher Exact Test utilized due to small cell sizes.

**Table 2 pharmacy-10-00160-t002:** Comparison of Providers and Pharmacists on Completion Rate, Revenue Generated, and Interventions During AWVs.

Variable	Providers; *N* = 139	Pharmacists; *N* = 116	*p*-Value
Initial AWV; *n* (%)	9 (6.5)	58 (50)	<0.0001 **
Subsequent AWV; *n* (%)	130 (93.5)	58 (50)	
AWVs Completed per Clinician	15.4	58	-
Total Revenue Generated	$34,308.94	$30,238.21	-
Revenue Generated per Clinician	$3812.10	$15,119.11	-
Potential Interventions Due *	488	393	0.356 **; −0.031 (−0.097, 0.035) †
Interventions Made * (% due at time of visit)	217 (44.5)	187 (47.6)	
Time with patient; mean ± SD	42 ± 13.5	36 ± 15.3	0.069 ‡

* Further breakdown in supplemental; ** Chi-square test was used to calculate the difference between interventions completed for providers vs. pharmacists. If a small *p*-value (*p*-value < 0.05) for the test occurs, this indicates that the null hypothesis of equal proportions can be rejected and that the proportions are unequal; † We provide an estimate of the difference in probability of interventions made as well as a confidence interval (i.e., Risk difference (95% CI)). If the confidence limits do not include zero as a likely value of the population mean difference, the difference is significant at the 0.05 level; ‡ Two-sample T-test was used to test the differences for the time with patient averages.

**Table 3 pharmacy-10-00160-t003:** Results of Patient Annual Wellness Visit Satisfaction Survey.

Survey Questions ^+^	Patients Seen by Providers; *N* = 26 Median (Min-Max; Mode)	Patients Seen by Pharmacists; *N* = 66 Median (Min-Max; Mode)
Q1: How satisfied were you with the scheduling process?	4.6 (4–5; 5)	4.5 (4–5; 4)
Q2: How satisfied were you with the punctuality of front office staff?	4.5 (3–5; 5)	4.5 (3–5; 4)
Q3: How satisfied were you with the punctuality of nursing staff?	4.6 (4–5; 5)	-
Q4: How satisfied were you with the punctuality of the pharmacist or provider?	4.6 (4–5; 5)	4.6 (3–5; 5)
Q5: How satisfied were you with the knowledge of the pharmacist or provider?	4.7 (4–5; 5)	4.6 (4–5; 5)
Q6: How satisfied were you with the duration of your appointment?	4.5 (4–5; 4/5)	4.5 (3–5; 5)
Q7: How satisfied were you with the opportunity to ask questions to the pharmacist or provider?	4.5 (4–5; 5)	4.7 (4–5;5)
Q8: How satisfied were you with the resources provided during your visit?	4.5 (4–5; 5)	4.6 (4–5; 5)

^+^ Patients could respond with very unsatisfied (1), unsatisfied (2), neutral (3), satisfied (4), or very satisfied (5).

**Table 4 pharmacy-10-00160-t004:** Results of Provider and Nurse Annual Wellness Visit Attitude Survey.

Survey Questions	Providers; *N* = 17 Median (Min-Max; Mode)	Nurse/Medical Assistant; *N* = 17 Median (Min-Max; Mode)
Q1: I am likely to recommend my patients receive a AWV every 12 months †.	4 (3–5; 4)	5 (4–5; 5)
Q2: I see a clear benefit with conducting routine AWVs for my qualifying patients †.	3.5 (3–5; 2)	4 (3–5; 4)
Q3: Rank the following visits. (1 = most enjoyable, 3 = least enjoyable) *.	2.5 (1–3; 3)	2.5 (1–3; 3)

† Participants could respond with strongly disagree (0), disagree (1), neutral (2), agree (3), or strongly agree (4). * Participants ranked the following visit types: Hospital Follow-Up/Establish Care, Follow-Up/Established Patient, or Medicare Annual Wellness Visit.

## Data Availability

The data presented in this study are available in the main article and Appendix A.

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
