# Peer review of "Impact of Clinical Pharmacy Expansion within a Rural Federally Qualified Health Center through Implementation of Pharmacist-Led Medicare Annual Wellness Visits"

_pharmacy, 2022, doi:10.3390/pharmacy10060160_

Round 1
Reviewer 1 Report
I thoroughly enjoyed reading the manuscript and had no comments. Recommend that the manuscript be published in its current form.
Author Response
Thank you for completing the peer review of our manuscript. We appreciate your time and consideration.
Reviewer 2 Report
Thank you for the opportunity to review this article analysing the impact of clinical pharmacy expansion within a rural health center through the implementation of pharmacy-led annual visits. The topic, the improvement of health conditions in rural settings, is of interest especially to public health and primary care practitioners (in fact, I wonder whether the choice of a pharmacy journal is the most appropriate, but this is an assessment that the authors will make in case). The design of the article is quite simple, a retrospective cohort study of the population that was visited over a five-month period in a rural health centre. The objectives set are reasonable and the analyses conducted were appropriate, but I do find some major flaws, some of which are also discussed by the authors in the limitations of the study. Some are no longer modifiable, e.g. the time of administration of the survey, but it would be necessary for the authors to make it clearer that their results are of rather limited value: data are presented on the higher number of visits by pharmacists compared to clinicians, but what value does this have beyond the economic factor? How did it translate into better health for users? Less time was spent with patients, which may mean less quality of service in the long term. Patient satisfaction after a visit is a fairly weak quality parameter: the study should have planned a long follow-up to seriously compare the quality of care provided by pharmacists and clinicians, or at least openly state the limitations in drawing conclusions from this data. Overall, when reading the work, one might have the impression of reading a 'promotional' publication on the employment of pharmacists in this centre. We do not wish to deny the benefit this may bring, but these benefits must be studied rigorously, or conclusions that are too far-fetched should be withdrawn.
Author Response
Thank you for completing the peer review of our manuscript. We appreciate your time and have detailed the improvements made to the manuscript below.
Recommendation was made to make it clear that our results are of site-specific value with limits to application and that benefits of the implementation must be studied rigorously. There was also a question of how did increased visits add to better health of patients. We have addressed these suggestions by adding to the conclusion by saying further evaluations are needed to determine the long-term impact on patients’ overall health (line 299). We believe that increased screenings and vaccines would translate into better overall health, but since this was not studied here we did not include this in our conclusion.
Since time with the patient was not considered significantly different between the groups, we did not find value in addressing this further (impact on quality of visit).
We also re-worded parts of our discussion to decrease the potentially “promotional” feel of the publication.
Thank you again for your time and consideration.
Reviewer 3 Report
This paper is well presented and provides findings that should encourage pharmacists to become further involved in Medicare annual wellness visits. The study is well conceived and written although some grammatical errors need attention. Importantly ethics approval was not included and as data have been collected by questionnaire this requires such approval.
Author Response
Thank you for completing the peer review of our manuscript. We appreciate your time and have addressed the grammatical errors and submitted our ethics approval for review.
Reviewer 4 Report
Overall, this is very well-written and easy for readers to understand. There were a few minor grammatical changes that are needed before publication.
I would like to see a breakdown of AWV done by nurses vs providers. My concern is that if most of the AWV listed under the provider category were actually done by nurses, then the economical incentive of using pharmacists is not valid as nurses are less expensive than pharmacists. If the provider visits were shared between nurses and providers than that is not as much a concern.
Please elaborate on incentives and how these figures were calculated.
Author Response
Thank you for completing the peer review of our manuscript. We appreciate your time and have addressed the grammatical errors.
Our nursing staff and providers worked jointly for visit (nurses completed intake and vitals, providers did pre-work, exam portion, and placed all orders for referrals).
We received incentive revenue reports from our study site’s quality department after the study period, a description of this has been added to our methods.
Thank you again for your time and consideration.
Round 2
Reviewer 2 Report
The authors have improved the manuscript in a satisfactory way.